# PhyBin: binning trees by topology

Ryan R. Newton[1] and Irene L.G. Newton[2]

[1] School of Informatics and Computing, Indiana University, Bloomington, IN, United States
[2] Department of Biology, Indiana University, Bloomington, IN, United States

## ABSTRACT

A major goal of many evolutionary analyses is to determine the true evolutionary history of an organism. Molecular methods that rely on the phylogenetic signal generated by a few to a handful of loci can be used to approximate the evolution of the entire organism but fall short of providing a global, genome-wide, perspective on evolutionary processes. Indeed, individual genes in a genome may have different evolutionary histories. Therefore, it is informative to analyze the number and kind of phylogenetic topologies found within an orthologous set of genes across a genome. Here we present PhyBin: a flexible program for clustering gene trees based on topological structure. PhyBin can generate *bins* of topologies corresponding to exactly identical trees or can utilize Robinson-Fould's distance matrices to generate *clusters* of similar trees, using a user-defined threshold. Additionally, PhyBin allows the user to adjust for potential noise in the dataset (as may be produced when comparing very closely related organisms) by pre-processing trees to collapse very short branches or those nodes not meeting a defined bootstrap threshold. As a test case, we generated individual trees based on an orthologous gene set from 10 *Wolbachia* species across four different supergroups (A–D) and utilized PhyBin to categorize the complete set of topologies produced from this dataset. Using this approach, we were able to show that although a single topology generally dominated the analysis, confirming the separation of the supergroups, many genes supported alternative evolutionary histories. Because PhyBin's output provides the user with lists of gene trees in each topological cluster, it can be used to explore potential reasons for discrepancies between phylogenies including homoplasies, long-branch attraction, or horizontal gene transfer events.

## INTRODUCTION

The advent of genomic sequencing has produced a large amount of data available for phylogenetic analysis and many researchers have attempted to utilize the phylogenetic signal found across the bacterial genome to develop species trees (*Daubin, Gouy & Perriere, 2001*; *Sicheritz-Ponten & Andersson, 2001*; *Daubin, Moran & Ochman, 2003*; *Bapteste et al., 2004*; *Zhaxybayeva et al., 2006*; *Ellegaard et al., 2013*). What has become clear from these analyses is that significant fractions of bacterial genomes do not follow the evolutionary history of their resident genome (*Bapteste et al., 2004*). These rogue genes are potentially undergoing evolutionary processes distinct from those felt by the rest of the

Corresponding author
Irene L.G. Newton,
irnewton@indiana.edu

resident genome or have arrived there via horizontal gene transfer events. In order, then, to understand the evolution of the genome, it would be useful to achieve an understanding of the evolution of each gene in the genome. Previous work by Sicheritz-Ponten and Andersson presented scripts combined the existing utilities BLAST, Clustalw, Paup 4.0* to provide a complete pipeline from genome to tree-binning analysis (*Sicheritz-Ponten & Andersson, 2001*). These kinds of complete solutions are convenient but constrain the user to the specific utilities chosen by the authors for alignment and phylogeny generation.

Here we present PhyBin, a computer program aimed at binning precomputed sets of non-reticulated trees in Newick format, a file format produced by the majority of tree building software. PhyBin is a utility rather than a complete solution; it can serve as a component in many genomics pipelines, and provides a useful addition to the landscape of tools for dissecting and visualizing large numbers of trees. After the user applies their chosen ortholog prediction and tree-building algorithms, PhyBin offers a quick way to visualize and browse the different evolutionary histories, either binned by topology and sorted by bin size, or in the form of a full hierarchical clustering based on Robinson-Foulds distance: i.e., a *tree of trees*.

## METHOD AND IMPLEMENTATION

### Generating orthologous sets and input trees

Genomic sequences were downloaded from NCBI Microbial Genome Projects. The *Wobachia* species complex is made up of several major clades, called supergroups, designated by alphabetical letters (*Baldo & Werren, 2007*). Accession numbers for the genomes analyzed here include: wUni and wVitA (wVitA: PRJDB1504; wUni: PRJNA33275), wBm (NC_006833.1), wPip-Pel (NC_010981.1), wHa (NC_021089.1), wRi (NC_012416.1), wMel (NC_002978.6), wNo (NC_021084.1), wAlbB (CAGB00000000.1), wBm (NC_006833.1), wOo (NC_018267.1). Orthologous gene sets were determined by Reciprocal Smallest Distance (RSD) algorithm (*Wall, Fraser & Hirsh, 2003*) with a $10^3$ cutoff for significance threshold and alignment length threshold of 80%. Orthologs were then aligned using ClustalW (*Larkin et al., 2007*) and ML trees were generated using RAxML (*Stamatakis, 2006*). The Newick format trees that resulted were used as input to PhyBin. The number of orthologous genes identified in this manner across all 10 taxa was 503.

## DESCRIPTION OF THE PROGRAM

PhyBin is a standalone command-line program, portable across all major operating systems (available at http://hackage.haskell.org/package/phybin). It runs in batch-mode and is easily usable from scripts. PhyBin has two major modes: it can run very quickly and classify identical tree topologies into bins, or it can compute the distance (*Robinson & Foulds, 1981*) between all pairs of trees and use that distance matrix to produce a configurable clustering of trees.

## Fast binning mode

The key algorithm PhyBin performs in this mode is tree normalization, computing a rooted, ordered *normal form* for all inputs (which are labeled, unrooted, unordered tree topologies). Previous work in this area has described a number of viable normal forms (*Chi, Yang & Muntz, 2005*). Conversion to a normal form ensures that all equivalent unrooted trees are converted into the same rooted tree, with a canonical root chosen. After conversion, the rooted trees are much faster to compare for equality than the unrooted trees would be, which enables fast binning.

PhyBin chooses the following strategy: it attempts to order subtrees by *weight* (number of tree nodes) and select the root node which is most balanced by weight (not depth)—that is, which minimizes the maximum weight of any child of the root. Node labels are used only to "break ties" between equally weighted subtrees, or equally balanced roots. Because input trees in Newick format are typically labeled only on the *leaves* (taxa), PhyBin generates labels for intermediate nodes in the tree by creating a set of all the leaves contained in that subtree, given a root to determine up/down direction. This set can be represented as a bit-vector and is also a key ingredient of computing Robinson-Foulds distance, which relies on identifying all such subsets (i.e., bipartitions induced by the tree). With labels for all nodes, equally weighted subtrees are ordered by label, and ties between potential roots are broken by comparing the labels of their children.

Once input trees are normalized, testing for equality of two trees is as simple as comparing their representation in memory (a single, linear traversal). Normalization itself appears expensive due to the cost of labeling interior nodes with all leaves under them ($O(N * I)$ for $N$ taxa and $I$ interior nodes), compounded by the fact that each intermediate node may have to consider each of its neighbors as a possible root and relabel itself $b$ times in a tree of maximum branching factor $b$, yielding an $O(N * I * b)$ asymptotic cost. However, in binning mode PhyBin runs much faster in the average case. One feature that enables PhyBin's efficiency is that it computes tree metadata—interior labels and "balanced" ratings—*lazily*, that is, on demand. Only when "tie breaking" is necessary between equally-weighted subtrees is an interior label computed at all. Likewise, only nodes "near the center" of the unrooted tree need to be considered for root status, those near the leaves need never be scored for balance.

After normalization, PhyBin performs binning, which amounts to inserting all normalized trees into a data structure *indexed* by tree topology. We define a total order over normalized trees (made possible by labels), and thereby represent the table of bins as a size-balanced binary tree supporting $O(\log(n))$ insertion times. A hash-table would be an alternative, but the tree representation allows us to insert trees into the table without evaluating (forcing) unnecessary interior labels in the normal forms, whereas hashing requires traversing the entirety of each normalized tree to compute its hash. When execution completes, the contents of each bin are written out to disk, in addition to a visualization of a representative average tree for that topology, computed by averaging branch lengths of the bin members.

## Pre-processing data

PhyBin helps users extract a clean dataset and detect problems with the data, such as trees with mismatching numbers of taxa. In order to facilitate comparisons across trees with different taxon names (i.e., gene names), PhyBin can extract portions of designations or use a separate table of rules for mapping genes to taxa. In addition, PhyBin can restrict its analyses to a subset of taxon, ignoring others (–prune).

A problem with the simple binning approach is that it is fragile to minor differences in trees caused by noise (e.g., short length branches with high variability). This becomes increasingly problematic with large numbers of taxa, especially when closely related taxa (different strains) are compared. Fortunately, a simple preprocessing step addresses this problem: PhyBin provides an option to collapse branches under two different conditions, a length threshold (for example, a length threshold of 0.01 would collapse all branches less than 0.01, in their place inserting a star topology) or a bootstrap support threshold (such that nodes with less than that threshold would be collapsed and the branch lengths from the taxa to the parent node would be added).

## Full clustering mode using Robinson-Foulds distance matrix

PhyBin reimplements the HashRF algorithm for full all-to-all Robinson Foulds distance (*Sul & Williams, 2007*), which is significantly faster than computing the distance matrix with repeated comparison of individual trees (e.g., PAUP (*Swofford & Sullivan, 2003*)). The HashRF algorithm is fast for today's data sizes (e.g., hundreds of taxa and thousands of trees), but it scales much more poorly than the basic binning algorithm at significantly larger sizes.

Because ortholog sets across different genomic comparisons will produce trees with different taxon memberships (as a result of paralogs or gene losses), a user may consider decomposing their trees with other software solutions (such as treeKO, (*Marcet-Houben & Gabaldon, 2011*)). Further, PhyBin is also capable of directly comparing these trees with different numbers of taxa using the leaf pruning method implemented in STRAW (*Shaw et al., 2013*). Specifically, in comparing trees with different taxa (–tolerant mode), the program first removes taxa that are not contained within each tree. If the taxon removed is in a polytomy, the parent and sister taxon are unchanged. However, in a binary node, taxon pruning would remove the intermediate node, retaining the branch lengths from the ancestor to the unpruned taxon. The –tolerant mode comes with a cost, however, as the more efficient HashRF algorithm cannot be used; instead Phybin falls back to the earlier PAUP-style algorithm.

A distance matrix alone is not directly useful for exploring the direct relationships between different gene trees. Thus, PhyBin uses the Robinson-Foulds distance matrix to compute a clustering of tree topologies, similar to the output of the simple binning mode, but able to identify trees that are merely *similar*, although not identical. A hierarchical clustering method is used. (If the user desires a different clustering method, they may use the distance matrix produced by PhyBin as input to a different processing pipeline.)

**Table 1 Compute time for PhyBin compared to two other distance matrix calculation programs.** The times below correspond to distance matrix computations only and were measured on the 150-taxa benchmark included with HashRF. All times in seconds. PhyBin times are given with different numbers of threads in parentheses. All times were taken on a 4-socket, 32-core Intel Xeon E7-4830 server running at 2.13 GHz with RHEL 6. Phylip was compiled with gcc 4.4.7 and "$-O_2$".

| Trees | PhyBin | HashRF | Phylip | DendroPy |
|-------|--------|--------|--------|----------|
| 100 | 0.269 | 0.056 | 22.1 | 12.8 |
| 1000 | 4.7 (1), 3.0 (2), 1.9 (4), 1.4 (8) | 1.7 | | |

With the hierarchical clustering method, there remain several clustering options to configure. The choice of clustering options can dramatically alter bin membership (Table S1), and running with several different options is a good way to get a sense for the range of possible outcomes. Specifically, the user may define the edit distance tolerated within clusters by providing a threshold, and may choose single, complete, or UPGMA linkage for clustering. Also if desired, rather than viewing a *flat* clustering of trees, the user may directly view a hierarchical clustering of the trees as a dendrogram. We believe PhyBin is the first program to date to provide this *tree-of-trees* output.

## Output formats

PhyBin is meant to be used in scripts and by other programs. Every output produced by PhyBin goes into a separate, simple text file—for example, the consensus tree for each cluster and the Robinson-Foulds distance matrix. Visualizations are produced separately and automatically in PDF files.

## Performance

There are very large differences in performance between existing programs for computing Robinson-Fould's distance matrices. The fundamental data-structures in this problem domain are sets and finite maps, for which there are many alternate representations (bit vectors, hash tables, balanced trees, etc.), providing a large space of possible implementations to explore. The sharpest contrast is between those programs that directly compare individual pairs of trees (PAUP, DendroPy), vs. those that insert all tree's bipartitions into a global structure and summarize it as a separate phase (e.g., HashRF). The later approach achieves much better cache locality.

PhyBin is written in a very high level language, Haskell, which supports radical forms of optimization, including safe semi-automatic parallelism. PhyBin uses purely functional (immutable) data-structures for representing trees and their bipartitons; in particular it relies heavily on the balanced-tree implementations `Data.Map` and `Data.Set` from the standard library. Nevertheless, when computing a matrix for a 150-taxa, 100-tree test (Table 1), PhyBin is 82 times faster than Philip (ANSI C) and 47.5 times faster than DendroPy (Python). However, PhyBin is still slower than HashRF by a factor of 2.8X-4.8X. HashRF was the first implementation that introduced high-performance techniques for RF matrices, and it introduced the algorithm on which PhyBin's implementation is based.

**Table 2 The behavior of PhyBin on an example dataset from the *Wolbachia* genus using *binning* mode.** Using PhyBin in *binning mode* on the *Wolbachia* orthologous gene set (503 trees total) results in different size and number of bins depending on branch length threshold. The number of bins drops dramatically between a branch length threshold of 0 and 0.02, indicating a small amount of noise in the dataset due to the use of fairly similar taxa.

| Branch length threshold | Number of bins | Number of singletons | Size of largest bin |
|---|---|---|---|
| 0 | 222 | 149 | 16 |
| 0.01 | 175 | 129 | 133 |
| 0.02 | 95 | 68 | 201 |
| 0.03 | 61 | 40 | 172 |
| 0.04 | 48 | 29 | 161 |

Unfortunately, the more widely used software (PAUP, DendroPy, Philip, etc.), remains slow. HashRF, the currently available fast alternative, is delicate and must be used carefully (for example, an extra character of whitespace in the input file results in a segmentation fault with no error message in version 6.0.1). Additionally, because HashRF provides only the core RF-distance computation, other tools are required for a biologist to be able to derive any conclusions from the output.

As a final note on performance, PhyBin was straightforward to parallelize (using our "LVar" parallelism library: *Kuper et al., in press*) and achieves a 2.54X parallel speedup at four cores, and peaks at a 3.11X speedup at eight cores, making it a bit faster than HashRF on our target platform (Table 1). Future work will focus on reducing contention on shared data structures to improve scaling.

## RESULTS AND DISCUSSION

We used PhyBin to identify how many phylogenies within the *Wolbachia* orthologous gene set support the supergroup divisions proposed by multi-locus sequence typing (*Baldo & Werren, 2007*). For comparative purposes in this analysis, a phylogeny for these 10 taxa was created using the concatenated, orthologous gene set (Fig. 1A). In actuality, PhyBin does not require an expectation for tree topology and searches through tree space for distinct topological categories. As an illustration of PhyBin's ability to reduce the noise in a dataset produced by small branch lengths (i.e., closely related taxa), we used the program in *binning mode* on the set of *Wolbachia* orthologs under increasing branch length thresholds (Table 2). We chose a threshold of 0.01 for our dataset as the average branch length over the entire set of validated trees was 0.04 with minimum and maximum branch lengths of 0 and 2.31, respectively. Using this threshold, in *binning* mode, the largest bin contains a topology that agrees with that of the published supergroups (133 members in largest bin, 175 total bins, Table 2, Fig. 1B). However, 174 other potential topologies exist in the dataset with 129 alternative topologies supported by only a single ortholog tree (Table 2).

In order to better explore this tree set, we took advantage of PhyBin's ability to generate a distance matrix for all trees. By calculating the Robinson-Foulds (RF) distance between all trees, we can better assess the differences between clusters in the tree dataset. For

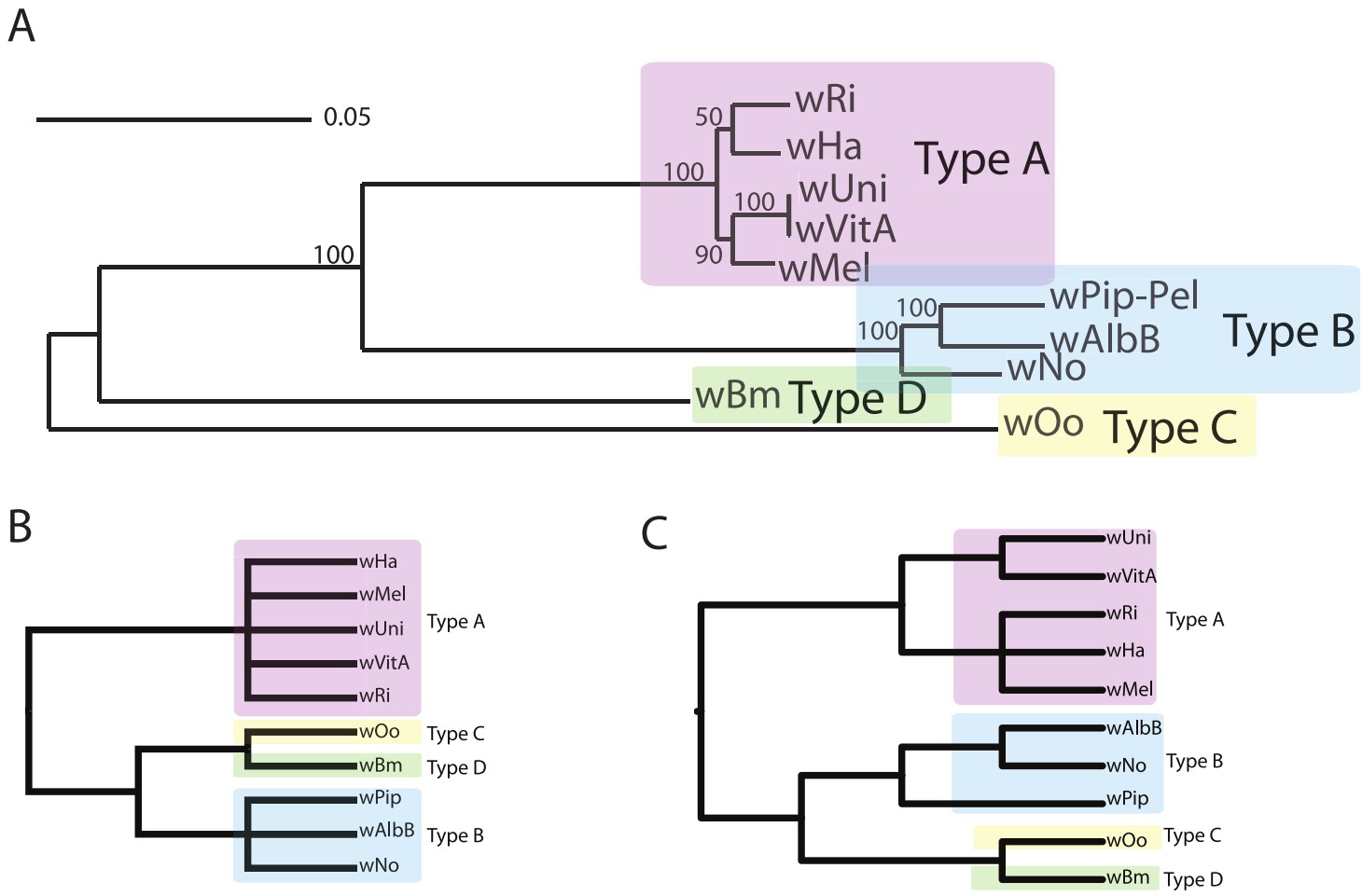

**Figure 1 Wolbachia supergroup trees produced by concatenation of a dataset of 508 orthologs or by PhyBin's binning and clustering algorithm.** In each of two modes (*full clustering* and *binning*) PhyBin is able to correctly recover the expected topology for the *Wolbachia pipientis* orthologs used herein. (A) Concatenated phylogeny based on 508 genes (using RAxML GTRGAMMA, bootstrap support based on 10,000 replicates). The four major supergroups are highlighted and denoted. (B) These same groups are recovered when PhyBin is run in either *binning* mode or (C) *full clustering* mode.

example, by increasing the RF-distance threshold to 2 and using the average-neighbor clustering algorithm to group our trees, the number of clusters drops dramatically to only 77 with the largest cluster containing a majority (72%) of genes. Again, this topology agrees with the published supergroup data and our result from the binning approach (Fig. 1C). Increasing the RF-distance threshold further provides increasing stringency in the detection of aberrant phylogenies – topologies not falling into the largest cluster at larger distance thresholds are likely to represent genes of interest in comparing evolutionary trajectories of these supergroups.

To test this hypothesis, we identified those *Wolbachia* genes that continue to display alternative evolutionary histories (that is, falling outside of the majority) even when clustering trees using increasingly large RF distances (Fig. 2B, Table 3). As expected, a large number of distinct topologies are not inconsistent with the supergroup clades (65

**Table 3** The behavior of PhyBin on an example dataset from the *Wolbachia* genus using full clustering mode. Using PhyBin in *full clustering mode* on the *Wolbachia* orthologous gene set (503 trees total) using average neighbor clustering produces a relatively small number of clusters, the largest comprised of a majority of orthologous genes.

| RF-distance threshold | Branch length cutoff | Number of clusters | Number of singletons | Size of largest cluster |
|---|---|---|---|---|
| 0 | n/a | 222 | 149 | 16 |
| 1 | n/a | 140 | 67 | 34 |
| 2 | n/a | 77 | 29 | 56 |
| 0 | 0.01 | 175 | 129 | 133 |
| 0 | 0.02 | 95 | 68 | 201 |
| 1 | 0.02 | 66 | 35 | 246 |

**Table 4** Wolbachia orthologs that do not conform to the dominant topology are highlighted by PhyBin. List of *Wolbachia* orthologous gene sets not conforming to the dominant topology when PhyBin is run using *full clustering mode* (–UPGMA, –editdist = 3). Protein products predicted to be secreted (based on screening using the Effective database (*Jehl, Arnold & Rattei, 2011*) are italicized.

| Topology group | Orthologs (using wMel designations) |
|---|---|
| Support for splitting group A | Major facilitator family transporter (WD0470) |
| | *Diaminopimelate epimerase (WD1208)* |
| | GTP cyclohydrolase (WD0003) |
| | *Metalopeptidase (WD0059)* |
| | *Periplasmic divalent cation tolerance (WD0828)* |
| | *RodA (WD1108)* |

distinct tree clusters do not support the major topology, using an RF-distance threshold of 1 and a branch length cutoff of 0.02, Table 3, Fig. 2B). We further investigated the ortholog set supporting the dissolution of supergroup A (Table 4). Interestingly, a majority of these orthologs are predicted to be secreted (using the Effective database predictions of sec signal or eukaryotic domains (*Jehl, Arnold & Rattei, 2011*), suggesting that perhaps interaction with the host would drive some of these orthologs in a different evolutionary direction compared to their resident genome. Another test of PhyBin's ability to detect orthologs under different evolutionary pressures would focus on the *Wolbachia* prophage, a mobile genetic element known to undergo horizontal transmission between strains (*Bordenstein & Wernegreen, 2004*; *Chafee et al., 2010*; *Kent & Bordenstein, 2010*; *Kent et al., 2011*). However, these phage orthologs do not occur across all of our 10 taxa included here and are therefore not suitable for testing support for the supergroups.

In conclusion, PhyBin is a new software program that efficiently and quickly groups phylogenies either by strict topological congruence or by clustering using RF distance. We believe that this tool, due to its ease of use, its speed, and informative output, will be of interest to evolutionary biologists and bioinformaticians alike.

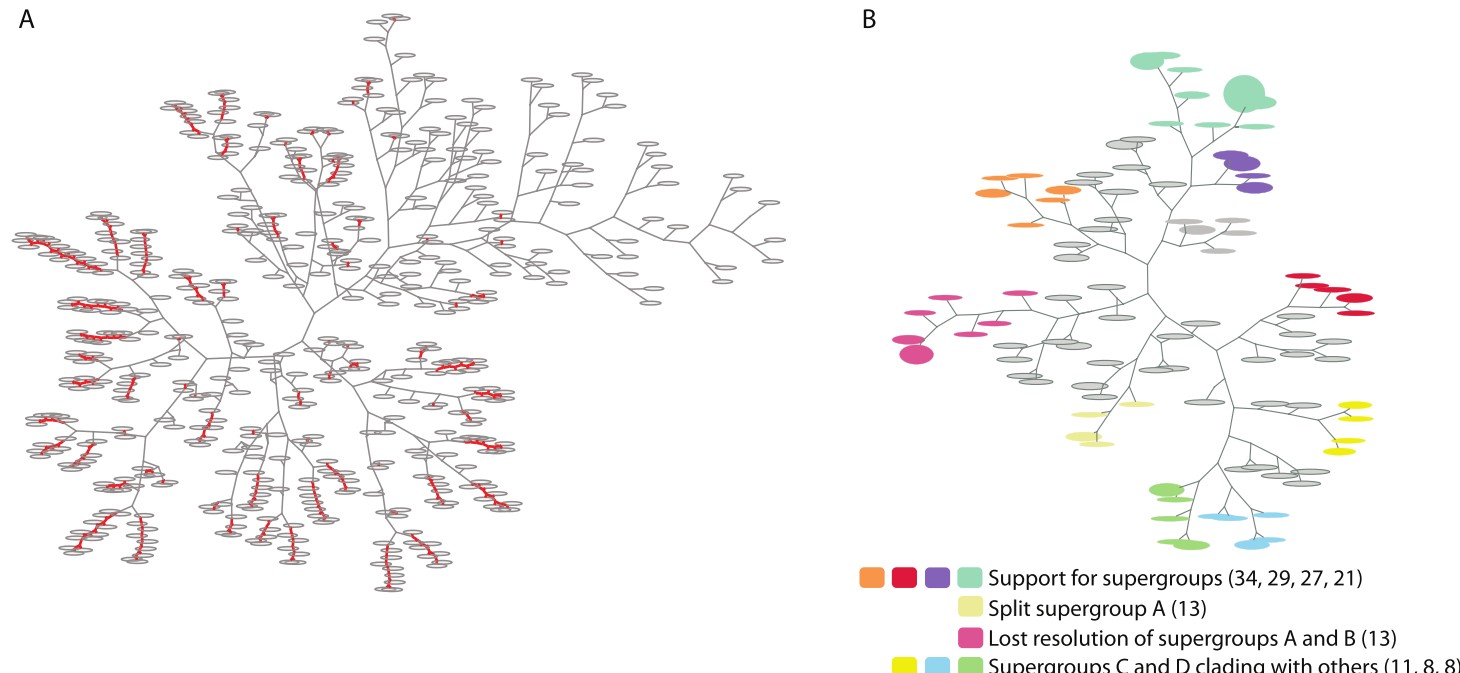

A B

Support for supergroups (34, 29, 27, 21)
Split supergroup A (13)
Lost resolution of supergroups A and B (13)
Supergroups C and D clading with others (11, 8, 8)

**Figure 2 Two trees of trees for the *Wolbachia* ortholog set as visualized by PhyBin.** Robinson-Foulds distance matricies produced by PhyBin are also visualized as a dendrogram by the software. (A) A *tree of trees* for the *Wolbachia* ortholog set (508 trees), clustered using an edit distance of 0, where identical topologies (nodes – grey ovals) are shown connected by a red line. Length of the branches connecting each node is proportional to the RF distance. (B) This dendogram is simplified by increasing the RF distance at which the trees are clustered (shown RF = 3). The top 10 clusters and their support different topologies are colored as indicated in the legend (with largest bin size for each cluster cluster in parentheses).

### Funding

Funding was provided by a generous laboratory startup given to ILGN by Indiana University. The funder had no role in study design, data collection and analysis, decision to publish, or preparation of the manuscript.

### Grant Disclosures

The following grant information was disclosed by the authors:
Indiana University.
NSF award CISE CCF: 1218375.

### Competing Interests

Irene Newton is an Academic Editor for PeerJ.

### Author Contributions

- Ryan R. Newton performed the experiments, contributed reagents/materials/analysis tools, wrote the paper.
- Irene L.G. Newton conceived and designed the experiments, performed the experiments, analyzed the data, wrote the paper.

## Data Deposition

DDBJ: wVitA: PRJDB1504; wUni: PRJNA33275.

## Supplemental Information

Supplemental information for this article can be found online at http://dx.doi.org/10.7717/peerj.187.

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
