# Peer review of "PhyBin: binning trees by topology"

_PeerJ, doi:10.7717/peerj.187_

## Round 0.1 · original submission · Major Revisions

The reviewers comments all seem reasonable to me and should be addressed. In particular I think anything you can add on comparing PhyBin to other tree clustering methods would be very useful.

·

Basic reporting

The paper presents a new utility to bin trees according to their topologies. They showcase the methods with an example of an analysis of Wolbachia gene tree sets. The method is publicly available and worked properly in my hands.

It is unclear to me whay the method includes a step of tree rooting. In fact RF distance can be computed on unrooted trees. Is this just a matter of computing optimization? is it possible that two topologies are rendered different solely because of a different root is used?. These issues should be clarified.

Similarly, Does the method enable trees including multifurcations as an input?

The paper reports the number of bins in their Wolbachia example, and mention that there is one majoritary topology, but what fraction of trees fall within the majoritary topology is not mentioned. It would be useful to provide the data of how many trees fall in each bin.

Experimental design

Rather than a primary research article, this is a methodological article. Although the method does not enable any fundamentally new type of analyses (such type of analyses could be done by concatenating a tree comparison tool, building a matrix and clustering), I think that the tool will be of interest to many users that want a simple and fast solution to the problem of tree-binning.

The tool enables collapsing clades based on branch lenghts. This is definitely useful, but I wonder why the method does not enable collapsing trees based on clade support values (e.g. bootstrap). That I guess would be a general need among potential users.

One clear limitation of the method is that it is only meant to compare trees that share identical number of labels, and contain one-to-one orthologs. As exemplified by the ~500 trees in their Wolbachia dataset this is a minimal fraction of the entire topological diversity (trees with paralogs, or missing genes are not included). Perhaps the authors could comment that this would be solvable by decomposing trees with duplication nodes with solutions such as the treeKO algorithm (Marcet-Houben et. al. http://www.ncbi.nlm.nih.gov/pubmed/21335609).

Validity of the findings

The paper includes an example of using the program on real data. The biological implications of the observed binning ("interaction with the host drives alternative evolution pattern) are not really highly supported and would need further work, but I think the example works fine to show the potential use of the utility.

It is somehow a pity that the comparison with other methodologies is restricted to the speed of RF computation, rather than with the bining itself. An idea is perhaps to compare how bins derived from PhyBin compare to other tree clusters (e,.g. Puigbo et. al. http://www.ncbi.nlm.nih.gov/pmc/articles/PMC3123530/) perhaps in other, already published datasets).

·

Basic reporting

This is a short and straightforward manuscript about a potentially useful tree binning/clustering tool. If the technical issues I encountered (described below) are dealt with, I would support the publication of this manuscript.

The authors acknowledge that they aren't the only people to have tried to deal with tree clustering, and they adequately explain the motivation for why their tool was created.

In terms of language, the only problem I encountered was the ungrammatical "can utilized Robinson-Fould's" in the abstract.

Experimental design

As this manuscript describes a bioinformatic tool, there isn't an experimental design as such, but they do give a test case showing an application of tree binning in an Wolbachia orthologous gene set.

Validity of the findings

While I have not tested the speed of the algorithm myself, it did seem to run quickly (on the data where I ran it successfully; see below), so the findings that their tool runs faster than most others seems plausible.

Additional comments

I initially tried to run the binary version of phybin 0.2.11 on Mac OSX 10.8.4. I had a set of 397 trees of orthologous genes from 7 deltaproteobacterial species that I thought would be a good test case. However, I encountered the error "openFile: resource exhausted (Too many open files)". I created a subset of 50 tree files and ran it again -- it managed to go further but finally died with "Waiting for asynchronous tasks to finish... phybin: Error running utility program: Error messages from neato:" (there were no messages after that line).

I did manage to successfully run the program on these trees on Linux, but the OSX version should be fixed. I also tried compiling the code myself with cabal but got the error: "Bio/Phylogeny/PhyBin.hs:45:18: Could not find module `Bio.Phylogeny.PhyBin.PreProcessor'", but I confess I know little about Haskell and this might have an obvious solution. Some tips for compilation on the homepage might help.

---

## Round 0.2 · accepted · Accept

I have reviewed your revisions and also sent the paper out to the prior reviewers. One of the reviewers responded and believes your revisions are acceptable. The other reviewer has not responded. As I and one of the reviews believe your revisions make the paper acceptable AND as I believe it is unnecessary to get the other reviewers comments, I am going to recommend accepting it.

·

Basic reporting

I have compiled and run the new version of phybin successfully on my test set. As my primary concerns in the previous version were technical, I believe they have now been addressed.

Experimental design

No further comments in this section.

Validity of the findings

The additional supplementary table on which compares nearest, average, and furthest neighbor on the Wolbachia dataset is a good addition to the manuscript